# Efficient and Parsimonious Agnostic Active Learning

**Tzu-Kuo Huang**
Microsoft Research, NYC
tkhuang@microsoft.com

**Alekh Agarwal**
Microsoft Research, NYC
alekha@microsoft.com

**Daniel Hsu**
Columbia University
djhsu@cs.columbia.edu

**John Langford**
Microsoft Research, NYC
jcl@microsoft.com

**Robert E. Schapire**
Microsoft Research, NYC
schapire@microsoft.com

## Abstract

We develop a new active learning algorithm for the streaming setting satisfying three important properties: 1) It provably works for any classifier representation and classification problem including those with severe noise. 2) It is efficiently implementable with an ERM oracle. 3) It is more aggressive than all previous approaches satisfying 1 and 2. To do this, we create an algorithm based on a newly defined optimization problem and analyze it. We also conduct the first experimental analysis of all efficient agnostic active learning algorithms, evaluating their strengths and weaknesses in different settings.

## 1 Introduction

Given a label budget, what is the best way to learn a classifier?

Active learning approaches to this question are known to yield exponential improvements over supervised learning under strong assumptions [7]. Under much weaker assumptions, streaming-based agnostic active learning [2, 4, 5, 9, 18] is particularly appealing since it is known to work for *any* classifier representation and *any* label distribution with an i.i.d. data source.[1] Here, a learning algorithm decides for each unlabeled example in sequence whether or not to request a label, never revisiting this decision. Restated then: What is the best possible active learning algorithm which works for *any* classifier representation, *any* label distribution, and is computationally tractable?

Computational tractability is a critical concern, because most known algorithms for this setting [e.g., 2, 16, 18] require explicit enumeration of classifiers, implying exponentially-worse computational complexity compared to typical supervised learning algorithms. Active learning algorithms based on empirical risk minimization (ERM) oracles [4, 5, 13] can overcome this intractability by using passive classification algorithms as the oracle to achieve a computationally acceptable solution.

Achieving generality, robustness, and acceptable computation has a cost. For the above methods [4, 5, 13], a label is requested on nearly *every* unlabeled example where two empirically good classifiers disagree. This results in a poor label complexity, well short of information-theoretic limits [6] even for general robust solutions [18]. Until now.

In Section 3, we design a new algorithm called ACTIVE COVER (AC) for constructing query probability functions that minimize the probability of querying inside the *disagreement region*—the set of points where good classifiers disagree—and never query otherwise. This requires a new algorithm that maintains a parsimonious cover of the set of empirically good classifiers. The cover is a result of solving an optimization problem (in Section 4) specifying the properties of a desirable

query probability function. The cover size provides a practical knob between computation and label complexity, as demonstrated by the complexity analysis we present in Section 4.

Also in Section 3, we prove that AC effectively maintains a set of good classifiers, achieves good generalization error, and has a label complexity bound tighter than previous approaches. The label complexity bound depends on the disagreement coefficient [10], which does not completely capture the advantage of the algorithm. In the end of Section 3 we provide an example of a hard active learning problem where AC is substantially superior to previous tractable approaches. Together, these results show that AC is better and sometimes substantially better in theory.

*Do agnostic active learning algorithms work in practice?* No previous works have addressed this question empirically. Doing so is important because analysis cannot reveal the degree to which existing classification algorithms effectively provide an ERM oracle. We conduct an extensive study in Section 5 by simulating the interaction of the active learning algorithm with a streaming supervised dataset. Results on a wide array of datasets show that agnostic active learning typically outperforms passive learning, and the magnitude of improvement depends on how carefully the active learning hyper-parameters are chosen.

More details (theory, proofs and empirical evaluation) are in the long version of this paper [14].

## 2 Preliminaries

Let $\mathbb{P}$ be a distribution over $\mathcal{X} \times \{\pm 1\}$, and let $\mathcal{H} \subseteq \{\pm 1\}^{\mathcal{X}}$ be a set of binary classifiers, which we assume is finite for simplicity.[2] Let $\mathbb{E}_X[\cdot]$ denote expectation with respect to $X \sim \mathbb{P}_{\mathcal{X}}$, the marginal of $\mathbb{P}$ over $\mathcal{X}$. The *expected error* of a classifier $h \in \mathcal{H}$ is $\mathrm{err}(h) := \mathrm{Pr}_{(X,Y)\sim\mathbb{P}}(h(X) \neq Y)$, and the error minimizer is denoted by $h^* := \arg\min_{h\in\mathcal{H}} \mathrm{err}(h)$. The *(importance weighted) empirical error* of $h \in \mathcal{H}$ on a multiset $S$ of importance weighted and labeled examples drawn from $\mathcal{X} \times \{\pm 1\} \times \mathbb{R}_+$ is $\mathrm{err}(h, S) := \sum_{(x,y,w)\in S} w \cdot \mathbb{1}(h(x) \neq y)/|S|$. The *disagreement region* for a subset of classifiers $A \subseteq \mathcal{H}$ is $\mathrm{DIS}(A) := \{x \in \mathcal{X} \mid \exists h, h' \in A \text{ such that } h(x) \neq h'(x)\}$. The *regret* of a classifier $h \in \mathcal{H}$ relative to another $h' \in \mathcal{H}$ is $\mathrm{reg}(h, h') := \mathrm{err}(h) - \mathrm{err}(h')$, and the analogous empirical regret on $S$ is $\mathrm{reg}(h, h', S) := \mathrm{err}(h, S) - \mathrm{err}(h', S)$. When the second classifier $h'$ in (empirical) regret is omitted, it is taken to be the (empirical) error minimizer in $\mathcal{H}$.

A streaming-based active learner receives i.i.d. labeled examples $(X_1, Y_1), (X_2, Y_2), \ldots$ from $\mathbb{P}$ one at a time; each label $Y_i$ is hidden unless the learner decides on the spot to query it. The goal is to produce a classifier $h \in \mathcal{H}$ with low error $\mathrm{err}(h)$, while querying as few labels as possible. In the IWAL framework [4], a decision whether or not to query a label is made *randomly*: the learner picks a probability $p \in [0, 1]$, and queries the label with that probability. Whenever $p > 0$, an unbiased error estimate can be produced using inverse probability weighting [12]. Specifically, for any classifier $h$, an unbiased estimator $E$ of $\mathrm{err}(h)$ based on $(X, Y) \sim \mathbb{P}$ and $p$ is as follows: if $Y$ is queried, then $E = \mathbb{1}(h(X) \neq Y)/p$; else, $E = 0$. It is easy to check that $\mathbb{E}(E) = \mathrm{err}(h)$. Thus, when the label is queried, we produce the importance weighted labeled example $(X, Y, 1/p)$.[3]

## 3 Algorithm and Statistical Guarantees

Our new algorithm, shown as Algorithm 1, breaks the example stream into epochs. The algorithm admits any epoch schedule so long as the epoch lengths satisfy $\tau_{m-1} \leq 2\tau_m$. For technical reasons, we always query the first 3 labels to kick-start the algorithm. At the start of epoch $m$, AC computes a *query probability function* $P_m: \mathcal{X} \to [0, 1]$ which will be used for sampling the data points to query during the epoch. This is done by maintaining a few objects of interest during each epoch in Step 4: (1) the best classifier $h_{m+1}$ on the sample $\tilde{Z}_m$ collected so far, where $\tilde{Z}_m$ has a mix of queried and predicted labels; (2) a radius $\Delta_m$, which is based on the level of concentration we want various empirical quantities to satisfy; and (3) the set $A_{m+1}$ consisting of all the classifiers with empirical regret at most $\Delta_m$ on $\tilde{Z}_m$. Within the epoch, $P_m$ determines the probability of querying an example in the disagreement region for this set $A_m$ of "good" classifiers; examples outside this

**Algorithm 1** ACTIVE COVER (AC)
---
**input:** Constants $c_1, c_2, c_3$, confidence $\delta$, error radius $\gamma$, parameters $\alpha, \beta, \xi$ for (OP), epoch schedule
$0 = \tau_0 < 3 = \tau_1 < \tau_2 < \tau_3 < \ldots < \tau_M$ satisfying $\tau_{m+1} \leq 2\tau_m$ for $m \geq 1$.

**initialize:** epoch $m = 0$, $\tilde{Z}_0 := \emptyset$, $\Delta_0 := c_1\sqrt{\epsilon_1} + c_2\epsilon_1 \log 3$, where $\epsilon_m := 32 \log(|\mathcal{H}|\tau_m/\delta)/\tau_m$.

1: Query the labels $\{Y_i\}_{i=1}^3$ of the first three unlabeled examples $\{X_i\}_{i=1}^3$, and set $A_1 := \mathcal{H}$,
$P_1 \equiv P_{\min,i} = 1$, and $S = \{(X_j, Y_j, 1)\}_{j=1}^3$.
2: **for** $i = 4, \ldots, n$, **do**
3:     **if** $i = \tau_m + 1$ **then**
4:        Set $\tilde{Z}_m = \tilde{Z}_{m-1} \cup S$, and $S = \emptyset$. Let

$$h_{m+1} := \arg\min_{h \in \mathcal{H}} \text{err}(h, \tilde{Z}_m), \quad \Delta_m := c_1\sqrt{\epsilon_m \text{err}(h_{m+1}, \tilde{Z}_m)} + c_2\epsilon_m \log\tau_m, \text{ and}$$

$$A_{m+1} := \{h \in \mathcal{H} \mid \text{err}(h, \tilde{Z}_m) - \text{err}(h_{m+1}, \tilde{Z}_m) \leq \gamma\Delta_m\}.$$

5:        Compute the solution $P_{m+1}(\cdot)$ to the problem (OP) and increment $m := m + 1$.
6:     **end if**
7:     **if** next unlabeled point $X_i \in D_m := \text{DIS}(A_m)$, **then**
8:        Toss coin with bias $P_m(X_i)$; add example $(X_i, Y_i, 1/P_m(X_i))$ to $S$ if outcome is heads,
otherwise add $(X_i, 1, 0)$ to $S$ (see Footnote 3).
9:     **else**
10:       Add example with predicted label $(X_i, h_m(X_i), 1)$ to $S$.
11:     **end if**
12: **end for**
13: Return $h_{M+1} := \arg\min_{h \in \mathcal{H}} \text{err}(h, \tilde{Z}_M)$.
---

region are not queried but given labels predicted by $h_m$ (so error estimates are not unbiased). AC computes $P_m$ by solving the optimization problem (OP), which is further discussed below.

The objective function of (OP) encourages small query probabilities in order to minimize the label complexity. The constraints (1) in (OP) bound the variance in our importance-weighted regret estimates for every $h \in \mathcal{H}$. This is key to ensuring good generalization as we will later use Bernstein-style bounds which rely on our random variables having a small variance. More specifically, the LHS of the constraints measures the variance in our empirical regret estimates for $h$, measured only on the examples in the disagreement region $D_m$. This is because the importance weights in the form of $1/P_m(X)$ are only applied to these examples; outside this region we use the predicted labels with an importance weight of 1. The RHS of the constraint consists of three terms. The first term ensures the feasibility of the problem, as $P(X) \equiv 1/(2\alpha^2)$ for $X \in D_m$ will always satisfy the constraints. The second empirical regret term makes the constraints easy to satisfy for bad hypotheses—this is crucial to rule out large label complexities in case there are bad hypotheses that disagree very often with $h_m$. A benefit of this is easily seen when $-h_m \in \mathcal{H}$, which might have a terrible regret, but would force a near-constant query probability on the disagreement region if $\beta = 0$. Finally, the third term will be on the same order as the second one for hypotheses in $A_m$, and is only included to capture the allowed level of slack in our constraints which will be exploited for the efficient implementation in Section 4. In addition to controlled variance, good concentration also requires the random variables of interest to be appropriately bounded. This is ensured through the constraints (2), which impose a minimum query probability on the disagreement region. Outside the disagreement region, we use the predicted label with an importance weight of 1, so that our estimates will always be bounded (albeit biased) in this region. Note that this optimization problem is written with respect to the marginal distribution of the data points $\mathbb{P}_X$, meaning that we might have infinitely many of the latter constraints. In Section 4, we describe how to solve this optimization problem efficiently, and using access to only unlabeled examples drawn from $\mathbb{P}_X$.

Algorithm 1 requires several input parameters, which must satisfy:

$$\alpha \geq 1, \ \xi \leq \frac{1}{8n\epsilon_M \log n}, \ \beta^2 \leq \frac{1}{\gamma n\epsilon_M \log n}, \ \gamma \geq 216, \ c_1 \geq 2\alpha\sqrt{6}, \ c_2 \geq 216c_1^2, \ c_3 \geq 1.$$

The first three parameters, $\alpha$, $\beta$ and $\xi$ control the tightness of the variance constraints (1). The next three parameters $\gamma, c_1$ and $c_2$ control the threshold that defines the set of empirically good classifiers; $c_3$ is used in the minimum probability (4) and can be simply set to 1.

---

**Optimization Problem** (OP) to compute $P_m$

$$\min_P \quad \mathbb{E}_X\left[\frac{1}{1 - P(X)}\right]$$

$$\text{s.t.} \quad \forall h \in \mathcal{H} \ \ \mathbb{E}_X\left[\frac{\mathbb{1}(h(x) \neq h_m(x) \wedge x \in D_m)}{P(X)}\right] \leq b_m(h), \tag{1}$$

$$\forall x \in \mathcal{X} \ \ 0 \leq P(x) \leq 1, \quad \text{and} \quad \forall x \in D_m \ \ P(x) \geq P_{\min,m} \tag{2}$$

$$\text{where} \quad \mathcal{I}_h^m(X) = \mathbb{1}(h(x) \neq h_m(x) \wedge x \in D_m),$$

$$b_m(h) = 2\alpha^2 \mathbb{E}_X[\mathcal{I}_h^m(X)] + 2\beta^2 \gamma \mathrm{reg}(h, h_m, \tilde{Z}_{m-1})\tau_{m-1}\Delta_{m-1} + \xi\tau_{m-1}\Delta_{m-1}^2, \tag{3}$$

$$P_{\min,m} = \min\left(\frac{c_3}{\sqrt{\frac{\tau_{m-1}\mathrm{err}(h_m, \tilde{Z}_{m-1})}{n\epsilon_M}} + \log \tau_{m-1}}, \frac{1}{2}\right). \tag{4}$$

---

*Epoch Schedules:* The algorithm takes an arbitrary epoch schedule subject to $\tau_m < \tau_{m+1} \leq 2\tau_m$. Two natural extremes are unit-length epochs, $\tau_m = m$, and doubling epochs, $\tau_{m+1} = 2\tau_m$. The main difference lies in the number of times (OP) is solved, which is a substantial computational consideration. Unless otherwise stated, we assume the doubling epoch schedule where the query probability and ERM classifier are recomputed only $\mathcal{O}(\log n)$ times.

**Generalization and Label Complexity.** We present guarantees on the generalization error and label complexity of Algorithm 1 assuming a solver for (OP), which we provide in the next section. Our first theorem provides a bound on generalization error. Define

$$\overline{\mathrm{err}}_m(h) := \frac{1}{\tau_m}\sum_{j=1}^m (\tau_j - \tau_{j-1})\mathbb{E}_{(X,Y)\sim\mathbb{P}}[\mathbb{1}(h(X) \neq Y \wedge X \in \mathrm{DIS}(A_j))],$$

$$\Delta_0^* := \Delta_0 \quad \text{and} \quad \Delta_m^* := c_1\sqrt{\epsilon_m \overline{\mathrm{err}}_m(h^*)} + c_2\epsilon_m \log \tau_m \ \text{ for } \ m \geq 1.$$

Essentially $\Delta_m^*$ is a population counterpart of the quantity $\Delta_m$ used in Algorithm 1, and crucially relies on $\overline{\mathrm{err}}_m(h^*)$, the true error of $h^*$ restricted to the disagreement region at epoch $m$. This quantity captures the inherent noisiness of the problem, and modulates the transition between $\mathcal{O}(1/\sqrt{n})$ to $\mathcal{O}(1/n)$ type error bounds as we see next.

**Theorem 1.** *Pick any* $0 < \delta < 1/e$ *such that* $|\mathcal{H}|/\delta > \sqrt{192}$. *Then recalling that* $h^* = \arg\min_{h \in \mathcal{H}} \mathrm{err}(h)$, *we have for all epochs* $m = 1, 2, \ldots, M$, *with probability at least* $1 - \delta$

$$\mathrm{reg}(h, h^*) \leq 16\gamma\Delta_m^* \quad \text{for all } h \in A_{m+1}, \quad \text{and} \tag{5}$$

$$\mathrm{reg}(h^*, h_{m+1}, \tilde{Z}_m) \leq 216\Delta_m. \tag{6}$$

The proof is in Section 7.2.2 of [14]. Since we use $\gamma \geq 216$, the bound (6) implies that $h^* \in A_m$ for all epochs $m$. This also maintains that all the predicted labels used by our algorithm are identical to those of $h^*$, since no disagreement amongst classifiers in $A_m$ was observed on those examples. This observation will be critical to our proofs, where we will exploit the fact that using labels predicted by $h^*$ instead of observed labels on certain examples only introduces a bias in favor of $h^*$, thereby ensuring that we never mistakenly drop the optimal classifier from $A_m$. The bound (5) shows that every classifier in $A_{m+1}$ has a small regret to $h^*$. Since the ERM classifier $h_{m+1}$ is always in $A_{m+1}$, this yields our main generalization error bound on the classifier $h_{\tau_m+1}$ output by Algorithm 1. Additionally, it also clarifies the definition of the sets $A_m$ as the set of good classifiers: these are classifiers which indeed have small population regret relative to $h^*$. In a realizable setting where $h^*$ has zero error, $\Delta_m^* = \tilde{\mathcal{O}}(1/\tau_m)$ leading to a $\tilde{\mathcal{O}}(1/n)$ regret after $n$ unlabeled examples are presented to the algorithm. On the other extreme, if $\overline{\mathrm{err}}_m(h^*)$ is a constant, then the regret is $\mathcal{O}(1/\sqrt{n})$. There are also interesting regimes in between, where $\mathrm{err}(h^*)$ might be a constant, but $\overline{\mathrm{err}}_m(h^*)$ measured

over the disagreement region decreases rapidly. More specifically, we show in Appendix E of [14] that the expected regret of the classifier returned by Algorithm 1 achieves the optimal rate [6] under the Tsybakov [17] noise condition.

Next, we provide a label complexity guarantee in terms of the disagreement coefficient [11]:
$\theta = \theta(h^*) := \sup_{r>0} \ \mathbb{P}_{\mathcal{X}}\{x \mid \exists h \in \mathcal{H} \text{ s.t. } h^*(x) \neq h(x), \mathbb{P}_{\mathcal{X}}\{x' \mid h(x') \neq h^*(x')\} \leq r\}/r$.

**Theorem 2.** *With probability at least* $1 - \delta$, *the number of label queries made by Algorithm 1 after* $n$ *examples over* $M$ *epochs is* $4\theta \, \overline{\text{err}}_M(h^*)n + \theta \cdot \tilde{\mathcal{O}}(\sqrt{n\overline{\text{err}}_M(h^*) \log(|\mathcal{H}|/\delta)} + \log(|\mathcal{H}|/\delta))$.

The theorem is proved in Appendix D of [14]. The first term of the label complexity bound is linear in the number of unlabeled examples, but can be quite small if $\theta$ is small, or if $\overline{\text{err}}_M(h^*) \approx 0$—it is indeed 0 in the realizable setting. The second term grows at most as $\tilde{\mathcal{O}}(\sqrt{n})$, but also becomes a constant for realizable problems. Consequently, we attain a logarithmic label complexity in the realizable setting. In noisy settings, our label complexity improves upon that of predecessors such as [5, 13]. Beygelzimer et al. [5] obtain a label complexity of $\theta\sqrt{n}$, exponentially worse for realizable problems. A related algorithm, Oracular CAL [13], has label complexity scaling with $\sqrt{n\text{err}(h^*)}$ but a worse dependence on $\theta$. In all comparisons the use of $\overline{\text{err}}_M(h^*)$ provides a qualitatively superior analysis to all previous results depending on $\text{err}(h^*)$ since this captures the fact that noisy labels outside the disagreement region do not affect the label complexity. Finally, as in our regret analysis, we show in Appendix E of [14] that the label complexity of Algorithm 1 achieves the information-theoretically lower bound [6] under Tsybakov's low-noise condition [17].

Section 4.2.2 of [14] gives an example where the label complexity of Algorithm 1 is significantly smaller than both IWAL and Oracular CAL by virtue of rarely querying in the disagreement region. The example considers a distribution and a classifier space with the following structure: (i) for most examples a single good classifier predicts differently from the remaining classifiers; (ii) on a few examples, half the classifiers predict one way and half the other. In the first case, little advantage is gained from a label because it provides evidence against only a single classifier. ACTIVE COVER queries over the disagreement region with a probability close to $P_{\min}$ in case (i) and probability 1 in case (ii), while others query with probability $\Omega(1)$ everywhere implying $\mathcal{O}(\sqrt{n})$ times more queries.

## 4 Efficient implementation

The computation of $h_m$ is an ERM operation, which can be performed efficiently whenever an efficient passive learner is available. However, several other hurdles remain. Testing for $x \in \text{DIS}(A_m)$ in the algorithm, as well as finding a solution to (OP) are considerably more challenging. The epoch schedule helps, but (OP) is still solved $\mathcal{O}(\log n)$ times, necessitating an extremely efficient solver.

Starting with the first issue, we follow Dasgupta et al. [9] who cleverly observed that $x \in D_m := \text{DIS}(A_m)$ can be efficiently determined using a single call to an ERM oracle. Specifically, to apply their method, we use the oracle to find[4] $h' = \arg\min\{\text{err}(h, \tilde{Z}_{m-1}) \mid h \in \mathcal{H}, h(x) \neq h_m(x)\}$. It can then be argued that $x \in D_m = \text{DIS}(A_m)$ if and only if the easily-measured regret of $h'$ (that is, $\text{reg}(h', h_m, \tilde{Z}_{m-1})$) is at most $\gamma\Delta_{m-1}$. Solving (OP) efficiently is a much bigger challenge because it is enormous: There is one variable $P(x)$ for every point $x \in \mathcal{X}$, one constraint (1) for each classifier $h$ and bound constraints (2) on $P(x)$ for every $x$. This leads to infinitely many variables and constraints, with an ERM oracle being the only computational primitive available.

We eliminate the bound constraints using barrier functions. Notice that the objective $\mathbb{E}_X[1/(1 - P(x))]$ is already a barrier at $P(x) = 1$. To enforce the lower bound (2), we modify the objective to

$$\mathbb{E}_X \left[ \frac{1}{1 - P(X)} \right] + \mu^2 \mathbb{E}_X \left[ \frac{\mathbb{1}(X \in D_m)}{P(X)} \right], \tag{7}$$

where $\mu$ is a parameter chosen momentarily to ensure $P(x) \geq P_{\min,m}$ for all $x \in D_m$. Thus, the modified goal is to minimize (7) over non-negative $P$ subject only to (1). We solve the problem in the dual where we have a large but finite number of optimization variables, and efficiently maximize the dual using coordinate ascent with access to an ERM oracle over $\mathcal{H}$. Let $\lambda_h \geq 0$ denote the

**Algorithm 2** Coordinate ascent algorithm to solve (OP)

---

**input** Accuracy parameter $\varepsilon > 0$. **initialize** $\boldsymbol{\lambda} \leftarrow \mathbf{0}$.

1: **loop**
2:     Rescale: $\boldsymbol{\lambda} \leftarrow s \cdot \boldsymbol{\lambda}$ where $s = \arg\max_{s \in [0,1]} \mathcal{D}(s \cdot \boldsymbol{\lambda})$.
3:     Find $\bar{h} = \arg\max_{h \in \mathcal{H}} \mathbb{E}_X \left[ \dfrac{\mathcal{I}_h^m(X)}{P_{\boldsymbol{\lambda}}(X)} \right] - b_m(h)$.
4:     **if** $\mathbb{E}_X \left[ \frac{\mathcal{I}_{\bar{h}}^m(X)}{P_{\boldsymbol{\lambda}}(X)} \right] - b_m(\bar{h}) \leq \varepsilon$ **then**
5:         **return** $\boldsymbol{\lambda}$
6:     **else**
7:         Update $\lambda_{\bar{h}}$ as $\lambda_{\bar{h}} \leftarrow \lambda_{\bar{h}} + 2 \dfrac{\mathbb{E}_X[\mathcal{I}_{\bar{h}}^m(X)/P_{\boldsymbol{\lambda}}(X)] - b_m(\bar{h})}{\mathbb{E}_X[\mathcal{I}_{\bar{h}}^m(X)/q_{\boldsymbol{\lambda}}(X)^3]}$.
8:     **end if**
9: **end loop**

---

Lagrange multiplier for the constraint (1) for classifier $h$. Then for any $\boldsymbol{\lambda}$, we can minimize the Lagrangian over each primal variable $P(X)$ yielding the solution

$$P_{\boldsymbol{\lambda}}(x) = \frac{\mathbb{1}(x \in D_m) q_{\boldsymbol{\lambda}}(x)}{1 + q_{\boldsymbol{\lambda}}(x)}, \quad \text{where } q_{\boldsymbol{\lambda}}(x) = \sqrt{\mu^2 + \sum_{h \in \mathcal{H}} \lambda_h \mathcal{I}_h^m(x)} \tag{8}$$

and $\mathcal{I}_h^m(x) = \mathbb{1}(h(x) \neq h_m(x) \wedge x \in D_m)$. Clearly, $\mu/(1+\mu) \leq P_{\boldsymbol{\lambda}}(x) \leq 1$ for all $x \in D_m$, so all the bound constraints (2) in (OP) are satisfied if we choose $\mu = 2P_{\min,m}$. Plugging the solution $P_{\boldsymbol{\lambda}}$ into the Lagrangian, we obtain the dual problem of maximizing the dual objective

$$\mathcal{D}(\boldsymbol{\lambda}) = \mathbb{E}_X \left[ \mathbb{1}(X \in D_m)(1 + q_{\boldsymbol{\lambda}}(X))^2 \right] - \sum_{h \in \mathcal{H}} \lambda_h b_m(h) + C_0 \tag{9}$$

over $\boldsymbol{\lambda} \geq 0$. The constant $C_0$ is equal to $1 - \Pr(D_m)$ where $\Pr(D_m) = \Pr(X \in D_m)$. An algorithm to approximately solve this problem is presented in Algorithm 2. The algorithm takes a parameter $\varepsilon > 0$ specifying the degree to which all of the constraints (1) are to be approximated. Since $\mathcal{D}$ is concave, the rescaling step can be solved using a straightforward numerical line search. The main implementation challenge is in finding the most violated constraint (Step 3). Fortunately, this step can be reduced to a single call to an ERM oracle. To see this, note that the constraint violation on classifier $h$ can be written as

$$\mathbb{E}_X \left[ \frac{\mathcal{I}_h^m(X)}{P(X)} \right] - b_m(h) = \mathbb{E}_X \left[ \mathbb{1}(X \in D_m) \left( \frac{1}{P(X)} - 2\alpha^2 \right) \mathbb{1}(h(X) \neq h_m(X)) \right]$$
$$- 2\beta^2 \gamma \tau_{m-1} \Delta_{m-1} (\text{err}(h, \tilde{Z}_{m-1}) - \text{err}(h_m, \tilde{Z}_{m-1})) - \xi \tau_{m-1} \Delta_{m-1}^2.$$

The second term of the right-hand expression is simply the scaled risk (classification error) of $h$ with respect to the actual labels. The first term is the risk of $h$ in predicting samples which have been labeled according to $h_m$ with importance weights of $1/P(x) - 2\alpha^2$ if $x \in D_m$ and 0 otherwise; note that these weights may be positive or negative. The last two terms do not depend on $h$. Thus, given access to $\mathbb{P}_{\mathcal{X}}$ (or samples approximating it, discussed shortly), the most violated constraint can be found by solving an ERM problem defined on the labeled samples in $\tilde{Z}_{m-1}$ and samples drawn from $\mathbb{P}_{\mathcal{X}}$ labeled by $h_m$, with appropriate importance weights detailed in Appendix F.1 of [14]. When all primal constraints are approximately satisfied, the algorithm stops. We have the following guarantee on the convergence of the algorithm.

**Theorem 3.** *When run on the $m$-th epoch, Algorithm 2 halts in at most $\Pr(D_m)/(8P_{\min,m}^3 \varepsilon^2)$ iterations and outputs a solution $\hat{\boldsymbol{\lambda}} \geq \mathbf{0}$ such that $P_{\hat{\boldsymbol{\lambda}}}$ satisfies the simple bound constraints in (2) exactly, the variance constraints in (1) up to an additive factor of $\varepsilon$, and*

$$\mathbb{E}_X \left[ \frac{1}{1 - P_{\hat{\boldsymbol{\lambda}}}(X)} \right] \leq \mathbb{E}_X \left[ \frac{1}{1 - P^*(X)} \right] + 4P_{\min,m}\Pr(D_m), \tag{10}$$

*where $P^*$ is the solution to (OP). Furthermore, $\|\hat{\boldsymbol{\lambda}}\|_1 \leq \Pr(D_m)/\varepsilon$.*

If $\varepsilon$ is set to $\xi^2 \tau_{m-1} \Delta_{m-1}^2$, an amount of constraint violation tolerable in our analysis, the number of iterations (hence the number of ERM oracle calls) in Theorem 3 is at most $\mathcal{O}(\tau_{m-1}^2)$. The proof is in Appendix F.2 of [14].

Table 1: Summary of performance metrics

| | OAC | IWAL$_0$ | IWAL$_1$ | ORA-OAC | ORA-IWAL$_0$ | ORA-IWAL$_1$ | PASSIVE |
|---|---|---|---|---|---|---|---|
| AUC-GAIN* | **0.151** | 0.150 | 0.142 | 0.125 | 0.115 | 0.121 | 0.095 |
| AUC-GAIN | 0.065 | **0.085** | 0.081 | 0.078 | 0.073 | 0.075 | 0.072 |

**Solving** (OP) **with expectation over samples:** So far we considered solving (OP) defined on the unlabeled data distribution $\mathbb{P}_{\mathcal{X}}$, which is unavailable in practice. A natural substitute for $\mathbb{P}_{\mathcal{X}}$ is an i.i.d. sample drawn from it. In Appendix F.3 of [14] we show that solving a properly-defined sample variant of (OP) leads to a solution to the original (OP) with similar guarantees as in Theorem 3.

## 5 Experiments with Agnostic Active Learning

While AC is efficient in the number of ERM oracle calls, it needs to store all past examples, resulting in large space complexity. As Theorem 3 suggests, the query probability function (8) may need as many as $\mathcal{O}(\tau_i^2)$ classifiers, further increasing storage demand. Aiming at scalable implementation, we consider an online approximation of AC, given in Section 6.1 of [14]. The main differences from AC are: (1) instead of a batch ERM oracle, it invokes an online oracle; and (2) instead of repeatedly solving (OP) from scratch, it maintains a fixed-size set of classifiers (and hence non-zero dual variables), called the *cover*, for representing the query probability, and updates the cover with every new example in a manner similar to the coordinate ascent algorithm for solving (OP). We conduct an empirical comparison of the following efficient agnostic active learning algorithms:

OAC: Online approximation of ACTIVE COVER (Algorithm 3 in Section 6.1 of [14]).
IWAL$_0$ and IWAL$_1$: The algorithm of [5] and a variant that uses a tighter threshold.
ORA-OAC, ORA-IWAL$_0$, and ORA-IWAL$_1$: Oracular-CAL [13] versions of OAC, IWAL$_0$ and IWAL$_1$.
PASSIVE: Passive learning on a labeled sub-sample drawn uniformly at random.

Details about these algorithms are in Section 6.2 of [14]. The high-level differences among these algorithms are best explained in the context of the disagreement region: OAC does importance-weighted querying of labels with an optimized query probability in the disagreement region, while using predicted labels outside; IWAL$_0$ and IWAL$_1$ maintain a non-zero minimum query probability everywhere; ORA-OAC, ORA-IWAL$_0$ and ORA-IWAL$_1$ query labels in their respective disagreement regions with probability 1, using predicted labels otherwise.

We implemented these algorithms in Vowpal Wabbit (`http://hunch.net/~vw/`), a fast learning system based on online convex optimization, using logistic regression as the ERM oracle. We performed experiments on 22 binary classification datasets with varying sizes ($10^3$ to $10^6$) and diverse feature characteristics. Details about the datasets are in Appendix G.1 of [14]. Our goal is to evaluate the test error improvement per label query achieved by different algorithms. To simulate the streaming setting, we randomly permuted the datasets, ran the active learning algorithms through the first 80% of data, and evaluated the learned classifiers on the remaining 20%. We repeated this process 9 times to reduce variance due to random permutation. For each active learning algorithm, we obtain the test error rates of classifiers trained at doubling numbers of label queries starting from 10 to 10240. Formally, let $\text{error}_{a,p}(d, j, q)$ denote the test error of the classifier returned by algorithm $a$ using hyper-parameter setting $p$ on the $j$-th permutation of dataset $d$ immediately after hitting the $q$-th label budget, $10 \cdot 2^{(q-1)}, 1 \leq q \leq 11$. Let $\text{query}_{a,p}(d, j, q)$ be the actual number of label queries made, which can be smaller than $10 \cdot 2^{(q-1)}$ when algorithm $a$ reaches the end of the training data before hitting that label budget. To evaluate an algorithm, we consider the area under its curve of test error against $\log$ number of label queries:

$$\text{AUC}_{a,p}(d, j) = \frac{1}{2} \sum_{q=1}^{10} \left( \text{error}_{a,p}(d, j, q+1) + \text{error}_{a,p}(d, j, q) \right) \cdot \left( \log_2 \frac{\text{query}_{a,p}(d, j, q+1)}{\text{query}_{a,p}(d, j, q)} \right).$$

A good active learning algorithm has a small value of AUC, which indicates that the test error decreases quickly as the number of label queries increases. We use a logarithmic scale for the number of label queries to focus on the performance with few label queries where active learning is the most relevant. More details about hyper-parameters are in Appendix G.2 of [14].

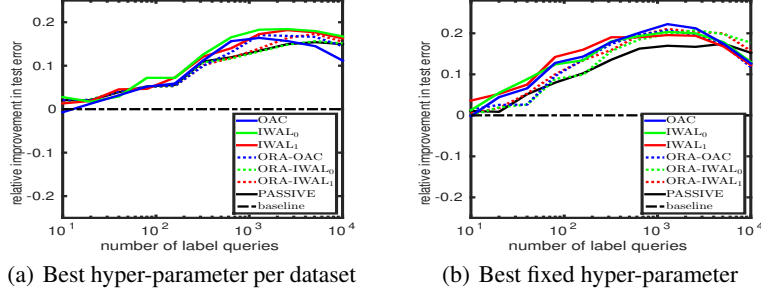

|                |                |
| :---: | :---: |
| (a) Best hyper-parameter per dataset | (b) Best fixed hyper-parameter |

Figure 1: Average relative improvement in test error v.s. number of label queries

We measure the performance of each algorithm $a$ by the following two aggregated metrics:

$$\text{AUC-GAIN}^*(a) \quad := \quad \operatorname*{mean}_{d} \operatorname*{max}_{p} \operatorname*{median}_{1 \le j \le 9} \left\{ \frac{\text{AUC}_{base}(d,j) - \text{AUC}_{a,p}(d,j)}{\text{AUC}_{base}(d,j)} \right\},$$

$$\text{AUC-GAIN}(a) \quad := \quad \operatorname*{max}_{p} \operatorname*{mean}_{d} \operatorname*{median}_{1 \le j \le 9} \left\{ \frac{\text{AUC}_{base}(d,j) - \text{AUC}_{a,p}(d,j)}{\text{AUC}_{base}(d,j)} \right\},$$

where $\text{AUC}_{base}$ denotes the AUC of PASSIVE using a default hyper-parameter setting, i.e., a learning rate of 0.4 (see Appendix G.2 of [14]). The first metric shows the maximal gain each algorithm achieves with the best hyper-parameter setting for each dataset, while the second shows the gain by using the single hyper-parameter setting that performs the best on average across datasets.

**Results and Discussions**. Table 1 gives a summary of the performances of different algorithms. When using hyper-parameters optimized on a per-dataset basis (top row in Table 1), OAC achieves the largest improvement over the PASSIVE baseline, with IWAL$_0$ achieving almost the same improvement and IWAL$_1$ improving slightly less. Oracular-CAL variants perform worse, but still do better than PASSIVE with the best learning rate for each dataset, which leads to an average of 9.5% improvement in AUC over the default learning rate. When using the best fixed hyper-parameter setting across all datasets (bottom row in Table 1), all active learning algorithms achieve less improvement compared with PASSIVE (7% improvement with the best fixed learning rate). In particular, OAC gets only 6.5% improvement. This suggests that careful tuning of hyper-parameters is critical for OAC and an important direction for future work.

Figure 1(a) describes the behaviors of different algorithms in more detail. For each algorithm $a$ we identify the best fixed hyper-parameter setting

$$p^* := \arg\max_{p} \operatorname*{mean}_{d} \operatorname*{median}_{1 \le j \le 9} \left\{ \frac{\text{AUC}_{base}(d,j) - \text{AUC}_{a,p}(d,j)}{\text{AUC}_{base}(d,j)} \right\}, \tag{11}$$

and plot the relative test error improvement by $a$ using $p^*$ averaged across all datasets at the 11 label budgets:

$$\left\{ \left( 10 \cdot 2^{(q-1)}, \operatorname*{mean}_{d} \operatorname*{median}_{1 \le j \le 9} \left\{ \frac{\text{error}_{base}(d,j,q) - \text{error}_{a,p^*}(d,j,q)}{\text{error}_{base}(d,j,q)} \right\} \right) \right\}_{q=1}^{11}. \tag{12}$$

All algorithms, including PASSIVE, perform similarly during the first few hundred label queries. IWAL$_0$ performs the best at label budgets larger than 80, while IWAL$_1$ does almost as well. ORA-OAC is the next best, followed by ORA-IWAL$_1$ and ORA-IWAL$_0$. OAC performs worse than PASSIVE except at label budgets between 320 and 1280. In Figure 1(b), we plot results obtained by each algorithm $a$ using the best hyper-parameter setting for each dataset $d$:

$$p_d^* := \arg\max_{p} \operatorname*{median}_{1 \le j \le 9} \left\{ \frac{\text{AUC}_{base}(d,j) - \text{AUC}_{a,p}(d,j)}{\text{AUC}_{base}(d,j)} \right\}. \tag{13}$$

As expected, all algorithms perform better, but OAC benefits the most from using the best hyper-parameter setting per dataset. Appendix G.3 of [14] gives more detailed results, including test error rates obtained by all algorithms at different label query budgets for individual datasets.

In sum, when using the best fixed hyper-parameter setting, IWAL$_0$ outperforms other algorithms. When using the best hyper-parameter setting tuned for each dataset, OAC and IWAL$_0$ perform equally well and better than other algorithms.

## Footnotes

[1]See the monograph of Hanneke [11] for an overview of the existing literature, including alternative settings where additional assumptions are placed on the data source (e.g., *separability*) [8, 3, 1].

[2]The assumption that $\mathcal{H}$ is finite can be relaxed to VC-classes using standard arguments.

[3]If the label is not queried, we produce an ignored example of weight zero; its only purpose is to maintain the correct count of querying opportunities. This ensures that $1/|S|$ is the correct normalization in $\mathrm{err}(h, S)$.

[4] See Appendix F of [15] for how to deal with one constraint with an *unconstrained* oracle.

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
