[Reviews · NeurIPS 2015]

Submitted by Assigned_Reviewer_1

Summary: This paper proposes a new agnostic active learning approach called Active Cover which only queries inside a region of data where classifiers disagree.

Quality: The authors show good results in the main text (but in the supplementary, Figures 6 and 7, the proposed method is often outperformed by other methods). Further, it is unclear how significant the differences between methods in Figure 2 of the main text are, i.e. how much of a practical differences does 0.01 error make? Some overall result would be useful (Figure 3 in the main paper is convincing as an overall result).

Originality: The idea behind this method seems novel (but this paper is outside my area of expertise).

Significance: This work seems significant (it tackles the important problem of stream-based agnostic active learning) and can have practical applications.

Clarity: The paper is not easy to follow by a non-expert. For example, the role and use of the parameters from L142 is not defined. In L146, what are unit-length and doubling epochs? In L158, what is meant by "population counterpart"? What does the D notation in step 2 of Algorithm 2 mean? In L358, what are "their respective disagreement regions"? How is the minimum test error computed? What is the computational complexity?
Summary: This paper proposes a novel approach for stream-based agnostic learning which seems to work well overall, but could be written more clearly.

Submitted by Assigned_Reviewer_2

The paper presents an active learning algorithm dedicated to the streaming setting: example per example, the algorithm decides whether to ask for a label or not. If good classifiers disagree on the example X, the proposed algorithm queries its label with a given probability given by the level of disagreement. If all good classifiers agree on X, the algorithm automatically labels the example with the predicted value.

I confess that I am unfamiliar with most of the current work in active learning, and that I did not verify the validity of the theorems (which proofs are provided in supplementary material). Nevertheless, it appears to be a substantial piece of work, which reunites several previous ideas and push them forward. The good behavior of the proposed algorithm is theoretically justified by generalization guarantees and a bound on the number of queries. Moreover, the empirical experiments corroborate the effectiveness of the approach.

The length and content of supplementary material is impressive (it contains 27 pages of non-trivial proofs, details about the experiments and supplementary results). The authors may have faced dilemmas when choosing information to give in the main paper (it is somewhat strange that the final learning algorithm is only explained in an appendix). This may explain the only complain I have about the paper: As a reader, I felt that the ideas were exposed in an expeditious manner. It would be appreciated to introduce some concepts that might be unknown to a reader not familiar with previous works in this area (like your humble reviewer!).

Specifically, the introduction should define the exact meaning that the following expressions have in the specific studied framework: "agnostic active learning", "classifier representation", "oracle". Similarly, Section 5 should precise the meaning of "Oracular-CAL".

Also, Algorithm 1 and Optimization Problem (OP) rely on many constant values, intervening in relatively complex expressions. To really understand their origin and meaning, the reader must go throughout the long proofs provided as supplementary material. I would appreciate being given an intuition of the meaning of each expression. Moreover, it would do no wrong to sometimes recall where expressions are defined. Indeed, the definitions are scattered. For example, D_j at Line 155 is defined inside Algorithm 1, \mathcal{D} and \mathcal{I}_h^m in Algorithm 2 (page 5) are respectively defined at Line 270 and by Equation (8) (page 6).

Minor typo in Appendix F : Theorem 2 => Theorem 3 (two times).
Summary: The proposed active learning algorithm is both mathematically grounded and empirically effective. I would vote more than 8 if the paper was easier to read.

Author Feedback
Author rebuttal: We thank all the reviewers for their comments and suggestions.
Our responses are as follows.

Reviewer 2:

1. As you observe, evaluating performance of different
methods across 23 datasets is not easy, and hence we mostly focus on aggregates in the main text. Figure 2 is meant to just give some intuitions and should not be overly read into. This will be clarified in the revision. There is an overall trend however, which is
captured in all of Figures 2, 6 and 7--namely OAC typically outperforms baselines at low query rates, which is the main point we are trying to make.

2. The various constants in line 142 are used in Algorithm 1. We will add some intuition about them in the revision and we give the key points below: \alpha controls the amount of variance we are willing to incur for good classifiers, while \beta captures the additional slack we can tolerate for bad classifiers. \xi controls the degree of approximation permissible in satisfying our constraints. \gamma, c_1 and c_2 all control the deviations between our empirical regret and regret estimates. In particular, they crucially control the expected regret of the classifier produced by our algorithm. \eta is introduced to simplify algebraic manipulation in the proof and can be set to an absolute numerical constant, as shown in line 142; c_3 is used in the minimum probability (3) and can be simply set to 1. We
will add explanations of these constants to the beginning of Section 3.

3. Unit-length epoch means every new example constitutes a new epoch. Doubling epochs mean every time the number of total examples seen so far doubles, a new epoch begins. They are defined in lines 146 (\tau_m = m) and 147 (\tau_{m+1} = 2\tau_m), respectively.

4. Population counterpart means that \Delta^* uses the true expectation (average over the whole population) in place of empirical average (over finite sample) in \Delta.

5. The \reg notations in Eq. (4) and (5) are defined in the beginning of Section 2, lines 93 and 94. The D notation in Step 2 of Algorithm 2 denotes the dual objective function, defined in Eq. (8).

6. More details about ORA-I and ORA-II are in Appendix G.2 of the supplementary material, which also explains what their disagreement regions are.

7. Regarding minimum test error: In Figures 2, 6 and 7, at each query rate we consider all hyper-parameter settings of an algorithm achieving that query rate, find their test errors, and take the minimum of them. The computational complexity is proportional to the
number of hyper-parameter settings.

Reviewers 3,4,5,6:
Thank you for your positive comments.

Reviewer 7:

Thank you for the positive feedback and for going through the details in the supplementary material. We will improve the clarity of the paper by adding more references to definitions of notations and
explanations of various expressions, such as the constants in Algorithm 1 and (OP) (Please see our response to Reviewer 2 above).